# Schizophrenia MEG Network Analysis Based on Kernel Granger Causality

**DOI:** 10.3390/e25071006

**Published:** 2023-06-30

**Authors:** Qiong Wang, Wenpo Yao, Dengxuan Bai, Wanyi Yi, Wei Yan, Jun Wang

**Affiliations:** 1School of Telecommunications and Information Engineering, Nanjing University of Posts and Telecommunications, Nanjing 210003, China; 2School of Physics and Information Engineering, Jiangsu Second Normal University, Nanjing 210013, China; 3Smart Health Big Data Analysis and Location Services Engineering Research Center of Jiangsu Province, School of Geographic and Biologic Information, Nanjing University of Posts and Telecommunications, Nanjing 210023, China; 4Department of Psychiatry, Affiliated Nanjing Brain Hospital, Nanjing Medical University, Nanjing 210029, China

**Keywords:** kernel Granger causality, effective network, schizophrenia MEG, nonequilibrium, complexity

## Abstract

Network analysis is an important approach to explore complex brain structures under different pathological and physiological conditions. In this paper, we employ the multivariate inhomogeneous polynomial kernel Granger causality (MKGC) to construct directed weighted networks to characterize schizophrenia magnetoencephalography (MEG). We first generate data based on coupled autoregressive processes to test the effectiveness of MKGC in comparison with the bivariate linear Granger causality and bivariate inhomogeneous polynomial kernel Granger causality. The test results suggest that MKGC outperforms the other two methods. Based on these results, we apply MKGC to construct effective connectivity networks of MEG for patients with schizophrenia (SCZs). We measure three network features, i.e., strength, nonequilibrium, and complexity, to characterize schizophrenia MEG. Our results suggest that MEG of the healthy controls (HCs) has a denser effective connectivity network than that of SCZs. The most significant difference in the in-connectivity strength is observed in the right frontal network (p=0.001). The strongest out-connectivity strength for all subjects occurs in the temporal area, with the most significant between-group difference in the left occipital area (p=0.0018). The total connectivity strength of the frontal, temporal, and occipital areas of HCs exhibits higher values compared with SCZs. The nonequilibrium feature over the whole brain of SCZs is significantly higher than that of the HCs (p=0.012); however, the results of Shannon entropy suggest that healthy MEG networks have higher complexity than schizophrenia networks. Overall, MKGC provides a reliable approach to construct MEG brain networks and characterize the network characteristics.

## 1. Introduction

Schizophrenia (SCZ) is a psychiatric disease characterized by significant impairments in the way reality is perceived, with delusions, hallucinations, and disorganized thinking and behavior [1]. In recent years, experts and scholars have held the view that SCZ may be related to aberrant brain network connections [2,3,4,5]. The human brain is a dynamic complex system, and each functional area does not work independently but achieves the purpose of completing a certain task through mutual coordination. The brain network can effectively quantify such coordinated interactions among various functional areas of the brain, making it particularly popular in elucidating complex brain dynamical structures. Complex network analysis [6,7,8,9], which could describe the brain as a network of nodes and edges, is a powerful tool to explore the characteristics of cerebral activity, especially in the brains of patients with psychiatric disorders [3,10,11]. With the development of neuroimaging, a thriving body of advanced techniques for understanding pathophysiological mechanisms has emerged, among which magnetoencephalography (MEG) stands out for its high temporal resolution and noninvasiveness [12,13]. A number of studies exploit MEG signals and brain network analysis to explore the dynamic features and underlying mechanisms of the SCZ brain. Houck et al. [14] utilized spatial independent component analysis and pairwise correlations between independent component timecourses to estimate the MEG network connectivity, and found that the patients with schizophrenia (SCZs) had abnormal connectivity within the frontal and temporal networks. Through assessing the characteristics of resting-state MEG networks using graph-theoretic analysis, Tagawa et al. [15] verified that the local networks of SCZs may disintegrate at both the microscale and macroscale levels, mainly in the beta band. Bai et al. [16] constructed the multiscale multidimensional recurrence plot for MEG signals of SCZs, and proved that the nonlinear dynamics of MEG signals in SCZs had lower predictability and laminarity. Lottman et al. [17] evaluated functional connectivity between resting-state MEG networks, and found that the delta band of SCZs revealed hypoconnectivity between sensorimotor and task-active networks.

Effective connectivity [18] is one of the approaches to construct physiological networks and represents the influence that one series exerts over another. It can provide directional and weighted information, which is helpful for tapping into the underlying structure of networks. Granger causality is a widely used measure of effective connectivity proposed by Granger [19] and has been used in economics [20], neuroscience [21], biology [22] and so on. When Granger causality was initially proposed, it was based on a linear model involving two variables. Considering the influence of other variables, conditional Granger causality [23] and partial Granger causality [24] have successively emerged. With the unremitting efforts of researchers, Granger causality has been generalized to the nonlinear analysis, namely kernel Granger causality [25,26,27], which is based on the theory of reproducing kernel Hilbert spaces, and performs linear relations in the feature space of suitable kernel functions by assuming an arbitrary degree of nonlinearity.

Network measures play an important role in exploring the characteristics and underlying structures of networks [6,7]. Weight [7,28] is an indispensable measure of an effective connectivity network, and the importance of a node can be measured by the in-, out- and total weight of a node. Moreover, for a complex and changeable physiological system, its structure is complex and constantly evolving, and the equilibrium and complexity measures of the network have the potential to better reveal the information interaction of the system. Equilibrium [29,30,31] refers to the constant performance of the statistical properties of a sequence over time or amplitude reversal. Entropy measures [32,33,34,35] are widely used to quantify the complexity of dynamical systems and can characterize the complexity of network information exchanges from an information-theoretic perspective.

In this paper, the MEG effective connectivity networks are constructed by multivariate inhomogeneous polynomial kernel Granger causality for SCZs and healthy controls (HCs). We adopt three measures (strength, nonequilibrium, and complexity) to explore the features of SCZ networks. The content is organized as follows: We describe the MEG data collection, Granger causality, and brain network measures in Section 2. A comparative test of multivariate inhomogeneous polynomial kernel Granger causality, bivariate linear Granger causality, and bivariate inhomogeneous polynomial kernel Granger causality is conducted in Section 3. Network analysis results are described in Section 4. The paper ends with a discussion in Section 5 and a brief conclusion in Section 6.

## 2. Materials and Methods

### 2.1. MEG Data

#### 2.1.1. Subjects

This experiment included 31 right-handed (self-reported) subjects, including 17 SCZs (4 women, age 25 ± 8.32) and 14 HCs (3 women, age 25.79 ± 5.29). Before the experiments, all subjects were fully informed and signed written informed consent. The Ethics Committee of Nanjing Brain Hospital approved this study. Subjects with a history of head trauma or drug abuse were excluded from the study. There were no significant between-group differences in age or sex.

#### 2.1.2. MEG Recording and Preprocessing

MEG recordings were obtained with a whole-head CTF MEG system with 275 channels (VSM Medical Technology Company, Coquitlam, BC, Canada), located in a magnetically shielded room at the Nanjing Brain Hospital. Prior to data acquisition, participants were instructed to remove all metallic wearables, lie in a supine position, remain awake but resting, and avoid blinking and making any eye or muscle movement. For each subject, MEG signals were recorded at a sampling frequency of 1200 Hz with a duration of 2 min. During the recording, if the participant was found to have made any movement that may have affected the accuracy of the result, the recorded signal was discarded and a new record made. All of the MEG datasets were preprocessed offline using the Fieldtrip toolbox [36] on MATLAB (version 2020b). The recordings containing artifacts were removed after manual checking and screening by experienced senior engineers. A bandpass filter of 0.1–200 Hz [37,38] was used to filter MEG data, and then direct current offset removal was achieved using a powerline filter (50 Hz and higher harmonics).

In our case, to avoid the multiple-comparison problems and computational limitations posed by this number of data, the 275 channels were grouped into five areas, namely the frontal (F), central (C), temporal (T), parietal (P), and occipital (O) areas. The temporal (T) area was divided into left and right regions, and the other four areas were divided into left, middle, and right regions. Consequently, the whole brain was divided into 14 regions, as illustrated in Figure 1.

### 2.2. Granger Causality

In this section, bivariate linear Granger causality (BLGC), bivariate inhomogeneous polynomial (IP) kernel Granger causality, and its multivariate pattern [25,26] are introduced.

#### 2.2.1. Bivariate Linear Granger Causality

Suppose that $X_{i}={\left(\xi_{i},\ldots ,\xi_{i+m-1}\right)}^{T}$, $Y_{i}={\left(\eta_{i},\ldots ,\eta_{i+m-1}\right)}^{T}$, and $x_{i}=\xi_{i+m}$ (for i=1,…,N) are treated as *N* realizations of the stochastic variables X, Y and x, respectively. X is an m×N matrix, where $X_{i}$ denotes the column, Z is a 2m×N matrix with vectors $Z_{i}={\left(X_{i}^{T},Y_{i}^{T}\right)}^{T}$, and x is a vector with elements $x_{i}$, i.e., $x={\left(x_{1},\ldots ,x_{N}\right)}^{T}$. For each component of X and Y, all values of the vector x have zero mean, and x is normalized, i.e., $x^{T}x=1$.

The vectors $\tilde{x}={\left({\tilde{x}}_{1},\ldots ,{\tilde{x}}_{N}\right)}^{T}$ and ${\tilde{x}}^{\prime }={\left({\tilde{x}}_{1}^{\prime },\ldots ,{\tilde{x}}_{N}^{\prime }\right)}^{T}$ are estimated by the linear regressions ${\tilde{x}}_{i}=\sum_{j=1}^{m}A_{j}\xi_{i+m-j}$ and ${\tilde{x}}_{i}^{\prime }=\sum_{j=1}^{m}A_{j}^{\prime }\xi_{i+m-j}+\sum_{j=1}^{m}B_{j}\eta_{i+m-j}$, respectively. $H\subseteq {ℜ}^{N}$ is the range of the matrix $K=X^{T}X$; then, $\tilde{x}$ is the projection of x on *H*. That is, if P is the projector on the space *H*, then $\tilde{x}=Px$. Analogously, $P^{\prime }$ is the projector on the 2m-dimensional space $H^{\prime }\subseteq {ℜ}^{N}$ with the range of the matrix $K^{\prime }=Z^{T}Z$; then, ${\tilde{x}}^{\prime }=P^{\prime }x$. $H^{\prime }$ can be decomposed as $H^{\prime }=H⊕H^{⊥}$, where $H^{⊥}$ is the space of all vectors of $H^{\prime }$ orthogonal to all vectors of *H*. $P^{⊥}$ is the projector on $H^{⊥}$. Suppose that y=x−Px; then, the linear Granger causality index can be denoted as [25]
(1)$$\begin{array}{c} \delta =\frac{{∥P^{⊥}y∥}^{2}}{1-{\tilde{x}}^{T}\tilde{x}} \end{array}$$
where $H^{⊥}$, which corresponds to the additional features due to the inclusion of {η} variables, is the range of the matrix $\tilde{K}=K^{\prime }-{\mathrm{PK}}^{\prime }-K^{\prime }P+{\mathrm{PK}}^{\prime }P$. Suppose that $H^{⊥}$ is spanned by the set of eigenvectors $\left\{t_{i}\right\}$, which are the eigenvectors with nonvanishing eigenvalues of $\tilde{K}$. Consequently, Equation (Equation 1) can be written as $\delta =\sum_{i=1}^{m}r_{i}^{2}$, where $r_{i}$ is the Pearson’s correlation coefficient of y and $t_{i}$. To avoid overfitting, Fisher’s r-to-z transformation and FDR correction are adopted to select the eigenvector $t_{i}$. Then, a filtered linear Granger causality index $\delta_{F}$ is obtained by summing only the values of $\left\{r_{i}\right\}$ that pass the FDR test:(2)$$\begin{array}{c} \delta_{F}=\sum_{i}r_{i^{\prime }}^{2} \end{array}$$
where $\delta_{F}$ measures the causality η→ξ. By exchanging the roles of the two time series, the causal interaction $\delta_{F}\left(\xi \to \eta \right)$ can be evaluated.

#### 2.2.2. Bivariate IP Kernel Granger Causality

The IP kernel of integer order *p* is $K_{p}\left(X,X^{\prime }\right)={\left(1+X^{T}X^{\prime }\right)}^{p}$. The bivariate IP Kernel Granger Causality (BKGC) is based on the theory of reproducing kernel Hilbert spaces [25,27]. In this case, $H\subseteq {ℜ}^{N}$ is the range of the Gram matrix K with elements $K_{ij}=K_{p}\left(X_{i},X_{j}\right)$ rather than the matrix $K=X^{T}X$ in the linear case. Analogously, the Gram matrix $K^{\prime }$ is organized with elements $K_{ij}^{\prime }=K\left(Z_{i},Z_{j}\right)$ instead of $K^{\prime }=Z^{T}Z$. Note that when p=1, it corresponds to the linear regression. Along the same line as described for the linear case, the BKGC only takes the eigenvectors of $\tilde{K}$ that pass the FDR test into account:(3)$$\begin{array}{c} \delta_{F}^{K}=\sum_{i^{\prime }}r_{i^{\prime }}^{2} \end{array}$$

#### 2.2.3. Multivariate IP Kernel Granger Causality

On the basis of BKGC, to assess the causality {x(a)}→{x(b)}, the Gram matrix K is evaluated as $K_{ij}=k\left(X_{i},X_{j}\right)$ with elements $X_{i}={\left(x{\left(1\right)}_{i}^{T},\ldots ,x{\left(M\right)}_{i}^{T}\right)}^{T}$ that contain all the input variables but those related to {x(a)}. The Gram matrix $K^{\prime }$ is then evaluated as $K_{ij}^{\prime }=k\left(Z_{i},Z_{j}\right)$ with elements $Z_{i}={\left(x{\left(1\right)}_{i}^{T},\ldots ,x{\left(a\right)}_{i}^{T},\ldots ,x{\left(M\right)}_{i}^{T}\right)}^{T}$ that contain all the input variables [25]. The target vector is then $x={\left(x{\left(b\right)}_{1+m},\ldots ,x{\left(b\right)}_{N+m}\right)}^{T}$. Consequently, in this case, the causality index of Multivariate IP Kernel Granger Causality (MKGC) is then calculated, as in Equation (Equation 3).

### 2.3. Brain Network Analysis

Complex network analysis is widely used for physiological and pathological data analysis [8,9]. Several complex network measures can be used to characterize directed weighted brain networks. A brief description of the complex network measures used in this study is presented in the following subsection.

#### 2.3.1. Weight

Given that *N* is the set of all nodes in the network, *W* is the weighted matrix of the network, and $w_{ij}$ is the weight of the link from node *i* to *j* [7], the in-connectivity strength of node *i* is the sum of all input weights of that node and is computed as $w_{i}^{in}=\sum_{j\in N}w_{ji}$. The out-connectivity strength is the sum of all output weights of the node and is computed as $w_{i}^{out}=\sum_{j\in N}w_{ij}$. The total connectivity strength of node *i*, referred to as the node strength $w_{i}$, consists of in-connectivity strength $w_{i}^{in}$ and out-connectivity strength $w_{i}^{out}$ and can be written as $w_{i}=w_{i}^{in}+w_{i}^{out}$. The information exchange of the whole network $w_{sum}=\sum_{i\in N}w_{i}^{in}=\sum_{i\in N}w_{i}^{out}$ provides information on the total level of weighted connectivity over the whole network.

#### 2.3.2. Network Nonequilibrium

The probability distributions of $w_{i}^{in}$ and $w_{i}^{out}$ of a node *i* are calculated as $p_{i}^{in}=w_{i}^{in}/w_{sum}$ and $p_{i}^{out}=w_{i}^{out}/w_{sum}$, respectively. The subtraction-based probabilistic difference parameter of in- and out-connectivity strength for a node *i* is formulated as [30,31]
(4)$$\begin{array}{c} Y_{S}=<p_{i}^{in},p_{i}^{out}>=p_{i}^{in}\frac{p_{i}^{in}-p_{i}^{out}}{p_{i}^{in}+p_{i}^{out}} \end{array}$$
where $p_{i}^{in}$ should not be smaller than $p_{i}^{out}$. If $p_{i}^{in}<p_{i}^{out}$, the roles of $p_{i}^{in}$ and $p_{i}^{out}$ should be exchanged, that is, $\langle p_{i}^{out},p_{i}^{in}\rangle$. If the in- or out-connectivity strength of a node does not exist, its corresponding probability distribution ($p_{i}^{in}$ or $p_{i}^{out}$) is 0. Cases such as this would lead to unreliable results for the division-based parameters, i.e., the relative entropy. However, we employ the subtraction-based parameters $Y_{S}$ in Equation (Equation 4) to avoid this situation.

The probability difference of in- and out-connectivity strength for a single node, called $Y_{S_{-}}L$, reflects the local nonequilibrium of the network, and we can assess the nonequilibrium of the whole network, called $Y_{S_{-}}W$, by summing the probability differences of each node.

#### 2.3.3. Complexity Measure

Shannon entropy [32] is usually considered fundamental and most natural when dealing with information content, and is typically used to evaluate the information content of a system by means of a probability distribution function. Given any arbitrary discrete probability distribution $P=(p_{i}:i=1\ldots ,M)$, Shannon’s logarithmic information measure [32,33,34,39] is $En=-\sum_{i=1}^{M}p_{i}\log p_{i}$. Shannon entropy measures the uncertainty and randomness in a given quantity of information and, as a measure of complexity, has gained popularity in nonlinear analysis [34,35]. In our work, $p_{i}^{in}$ and $p_{i}^{out}$ are the probability distributions of the in- and out-connectivity strength of a node *i*, respectively. Consequently, the Shannon entropy measures of in- and out-connectivity strength for networks are given as follows:(5)$$\begin{array}{c} \begin{array}{c} En\text{\_}in=-\sum_{i}p_{i}^{in}\log p_{i}^{in}\\ En\text{\_}out=-\sum_{i}p_{i}^{out}\log p_{i}^{out} \end{array} \end{array}$$

## 3. Model Data Tests

In this section, we construct models of five nonlinear time series using the coupled first-order autoregressive processes to verify the effectiveness of MKGC in comparison to BLGC and BKGC, as displayed in Equation (Equation 6):(6)$$\begin{array}{l} x_{1}\left(t\right)=\left(1-e\right)\left(1-ax_{1}^{2}\left(t-1\right)\right)+e\left(1-ax_{2}^{2}\left(t-1\right)\right)+s\tau_{1}\left(t\right)\\ x_{2}\left(t\right)=1-ax_{2}^{2}\left(t-1\right)+s\tau_{2}\left(t\right)\\ x_{3}\left(t\right)=\left(1-e\right)\left(1-ax_{3}^{2}\left(t-1\right)\right)+e\left(1-ax_{1}^{2}\left(t-1\right)\right)+s\tau_{3}\left(t\right)\\ x_{4}\left(t\right)=\left(1-e\right)\left(1-ax_{4}^{2}\left(t-1\right)\right)+e\left(1-ax_{1}^{2}\left(t-1\right)\right)+s\tau_{4}\left(t\right)\\ x_{5}\left(t\right)=\left(1-e\right)\left(1-ax_{5}^{2}\left(t-1\right)\right)+e\left(1-ax_{4}^{2}\left(t-1\right)\right)+s\tau_{5}\left(t\right) \end{array}$$
where e=0.2, a=1.8, s=0.02, and $\tau^{\prime }s$ are unit variance Gaussian noise terms. The causal relationships implemented in these equations are 1→3, 1→4, 2→1, and 4→5, and are illustrated in Figure 2a. Analyzing segments of length N=1000, we evaluate causality for all pairs of maps within the 200 simulation runs.

The model results for the three methods are shown in Figure 2b. The results of BLGC analysis show that the causality index of 1→3, 1→4, 2→1 and 4→5 is extremely low, with false causal influences for 2→3 and 2→4, which indicates that BLGC is not suitable for detecting the causal interaction of nonlinear time series. Both BKGC and MKGC analysis reveal the influences 1→3, 1→4, 2→1, and 4→5 with fairly large values. The slight causal interactions of 1→5, 2→3, 2→4, 3→4, 3→5 and 4→3, which are mediated by the other interactions, are only revealed by BKGC, while the MKGC indicates that they are nonsignificant (the causality index was equal to zero). The results suggest that bivariate evaluation has a limitation in that it cannot be used to discern whether the influence between two time series is direct or mediated by another one. MKGC not only has the ability to identify causal relationships between nonlinear time series but also distinguishes direct influences from indirect influences between time series.

To further analyze the validity of BLGC, BKGC and MKGC, the sensitivity, specificity, and Matthews correlation coefficient (*Mcc*) of the three methods are calculated. By comparing the rows of the target matrix with those of the actual matrix, we can obtain four parameters: true negative (*TN*), true positive (*TP*), false negative (*FN*), and false positive (*FP*). The sensitivity, specificity, and *Mcc* are evaluated as follows:(7)$$\begin{array}{c} Sensitivity(\%)=100\times TP/(TP+FN)\\ Specificity(\%)=100\times TN/(TN+FP)\\ Mcc\left(\%\right)=100\times \frac{TP\times TN-FP\times FN}{\sqrt{\left(TP+FP)(TP+FN)(TN+FP)(TN+FN\right)}} \end{array}$$

Table 1 quantitatively displays the performances of the average results from 200 runs with the parameters of sensitivity, specificity and *Mcc* with the three methods on the five time series above. Out of the three approaches, MKGC outperforms BLGC and BKGC.

In summary, compared with the other two methods, MKGC can more accurately identify pathways of causal interaction between nonlinear time series, thus providing a powerful tool for determining the effective connectivity between brain regions and for brain network construction.

## 4. Network Analysis on Schizophrenia MEG Data

For each individual brain dataset, the brain regions are defined as the nodes of the network, and the representative time series of each brain region is obtained by averaging the MEG time series across all channels within that brain region. We exploit MKGC to calculate the causal interactions at the regional level in MATLAB (version 2020b), then construct directed weighted networks for MEG activities and characterize the effects of SCZ on network interactions. The order *m* of MKGC for the MEG data is set to 1 using Bayesian information criterion (BIC) [40]. According to Marinazzo et al. [25], Liao et al. [26], and the result of the model data test above, we choose p=2 for the order of IP kernel. The Mann–Whitney U test is performed to find significant differences in the network parameters.

We adopt surrogate theory [41,42] to assess the statistical significance of effective connectivity constructed by MKGC. A total of 200 sets of surrogate data for each real dataset are generated using the improved amplitude-adjusted Fourier transform [43]. To detect whether the original MKGC connection is significantly different from the surrogate values, we calculate the statistic φ between the original and the mean surrogate value as $\varphi \equiv \frac{\left|Q_{D}-\mu_{H}\right|}{\sigma_{H}}$ according to J. Theiler [41]. The value p=0.05 is employed as the significance level, and the non-significant effective connectivity level is removed [42,44].

### 4.1. MEG Effective Connectivity Network

The group average effective connectivity networks of HCs and SCZs are presented in Figure 3a,b. It is obvious that HCs have more and denser interregional connections than SCZs. The average value of effective connectivity over the whole brain of SCZs (0.008 ± 0.002) is lower than that over the whole brain of HCs (0.009 ± 0.002). Moreover, the Mann–Whitney U test is performed to investigate the between-group differences in each effective connection, and the connections with significant differences (p<0.05) between HCs and SCZs are formed into the directed differential connectivity graph, as shown in Figure 3c. In total, 24 of all the effective connectivity values show significant differences, which mainly exist in the frontal temporal (FT) and occipital parietal (OP) subnetworks, among which ZC→ZP (p=0.001), ZP→LO (p=0.004), and ZF→LF (p=0.006) have more acceptable differences.

### 4.2. Network Connectivity Strength

The weights of the effective connectivity networks represent the strength of the causal interactions. Exploring the strength and their relationships is beneficial for comprehensively understanding the natural properties and underlying structures of the networks. To this end, we investigate the in-, out-, and total-connectivity strength of brain regions successively.

The maximum values of the in-connectivity strength are observed in ZF for both HCs (0.099) and SCZs (0.077). The in-connectivity strength of each brain region, excluding ZC and RC, of the HCs is greater than that of SCZs. Significantly reduced levels occur in the LF, ZF, RF, LT, RT, and LP regions of SCZs, among which RF exhibits the most significant difference (p=0.001). In addition, significant differences are most widely distributed in the frontal (LF, ZF, and RF) and temporal (LT and RT) regions. However, there are no significant differences in the central and occipital regions (Figure 4).

The strongest out-connectivity strength for all subjects occurs in the temporal area (including LT and RT), with out-connectivity strengths of RT for HCs and SCZs of 0.087 and 0.059, respectively, and out-connectivity strengths of LT for HCs and SCZs of 0.071 and 0.055. The out-connectivity strength of all regions of HCs is greater than that of SCZs, excluding RC and RP; the between-group differences are significant in the LF, ZF, ZP, LO, and ZO regions, among which LO exhibits the most significant difference (p=0.0018). Furthermore, significant differences are most widely distributed in the frontal (LF and ZF) and occipital (LO and ZO) regions, while no significant differences are observed in the temporal and central regions (Figure 5).

In another strength analysis, we focus on the total connectivity strength of brain regions. It is obvious that the total connectivity strength of brain regions is nonhomogeneous. Specifically, the total connectivity strength of the peripheral brain regions (LF, ZF, RF, LT, and RT) of the subjects is higher than that of the middle brain regions (LC, ZC, RC, LP, ZP, and RP). The total connectivity strength of the frontal (LF, ZF, and RF) and temporal (LT and RT) regions of all subjects exhibits higher values compared with other brain regions. Moreover, relative to that of the SCZs, the total connectivity strength is generally greater over the vast majority of the brain regions of HCs, and, in particular, significant differences in LO (p=0.004), LF (p=0.009), and RT (p=0.009) exist, as displayed in Figure 6.

### 4.3. Network Nonequilibrium

The difference between the in-connectivity and out-connectivity strengths in a brain region reflects the increase or decrease in the amount of information in the brain region. Measuring the parameter $Y_{S_{-}}L$ [30,31] for each brain region, we can explore the local nonequilibrium of the network. The maximum value of $Y_{S_{-}}L$ is observed in ZF for both HCs and SCZs, and is 0.167 for the HCs and 0.232 for the SCZs, but there is no significant difference. The value of $Y_{S_{-}}L$ in the frontal (ZF, RF, and LF) region of SCZs is higher than that in the other four brain regions. However, LF, RF, ZC, and ZP all have acceptable discriminations between the two groups, and ZC (p=0.005) has a better discrimination, as illustrated in Figure 7.

In addition, the parameter $Y_{S_{-}}W$ is employed to assess nonequilibrium features over the whole brain. The $Y_{S_{-}}W$ value of the MEG network constructed by the MKGC for the HCs is 0.741, and that of the SCZs is 0.937, which is significantly larger than that of the HCs (p=0.012), as displayed in Figure 8.

### 4.4. Network Complexity

Shannon entropy, one of the measures that characterize static complexity [34], can be exploited to express the amount of information in a network. We quantify the Shannon entropy of in-connectivity and out-connectivity strength for all brain regions. The Shannon entropy of the in-connectivity strength for HCs is 2.311 and that for SCZs is 2.227. No statistically significant difference (p=0.225) is found between SCZs and HCs (Figure 9a). In addition, the Shannon entropy of the out-connectivity strength for HCs is 2.257, and that for SCZs is 2.224. The two groups do not have acceptable discrimination (p=0.593) (Figure 9b).

From the above analysis, it can be seen that MKGC can effectively characterize the causal interactions between nonlinear time series. MKGC is a reliable tool to construct MEG networks for characterizing the physiological and pathological features of SCZs.

## 5. Discussion

The directed weighted networks based on MEG signals are constructed by exploiting MKGC to explore the dynamic structure and characteristics of schizophrenic brain networks. However, during the analysis, we found that there are some related issues that need further discussion.

From the above network strength analysis, it can be seen that the SCZ networks have lower values of in-, out- and total-connectivity strength in the frontal, temporal, occipital, and parietal regions, especially in the frontal regions. A wealth of SCZ studies on structural, functional and effective connectivity [4,45,46,47] hold similar opinions and have reported that a decreased level of connectivity was observed in SCZ networks. In addition, the frontal lobe is closely related to cognitive functions, and the abnormal performance of the frontal lobe indicates cognitive dysfunction in SCZs. Numerous neuroimaging studies have revealed that the frontal lobe exhibits abnormal performance in SCZs, for instance, an increased count of pathological myelinated fibers [48], reduced centrality [49], altered clustering [50], and longer path lengths [51]. In our findings, the network connectivity strength and local nonequilibrium of the frontal regions in SCZ networks are significantly different to those in healthy networks, which is consistent with the previous statements.

We then discuss the contradictory findings about the Shannon entropy and nonequilibrium measures. The nonequilibrium in the SCZs’ network is significantly larger than that in the HCs’ network, while the Shannon entropy of in-connectivity and out-connectivity in the SCZs’ networks is smaller than that in the HCs’ networks. The mathematical reason for this contradiction lies in the fact that nonequilibrium and Shannon entropy measure probabilistic distributions from different perspectives. The smaller the probabilistic differences, the smaller the nonequilibrium but the larger the entropy. Moreover, these contradictory results suggest that the nonequilibrium and entropy approaches, as information-theoretic approaches, target different aspects of complex systems. Shannon entropy characterizes complexity [34,35], whereas nonequilibrium [30,31] describes the features of the in–out fluctuations in network interactions. This comparative study at different levels for the characteristics of complex systems encourages us to better grasp features of the system from a broader perspective and have a deeper understanding of the specific properties of systems.

Another issue concerns the quantitative nonequilibrium for network interactions. Previous studies [52,53] have described network time irreversibility. It is important to note the differences between the concepts of network nonequilibrium and time irreversibility. Time reversibility describes the property whereby a process is invariant under the reverse time scale, which is an important approach to measure the characteristics of nonequilibrium. Statistically, time irreversibility can be measured by the probabilistic differences between forward and backward series or those between symmetric vectors. However, once a network is constructed, it is impossible to use the forward and reverse processes of time irreversibility to detect probability differences. Consequently, the research on the time irreversibility of a network essentially aims to measure a kind of nonequilibrium from a different perspective. Moreover, relative to time irreversibility, visibility graphs [52], fluctuations in the vector distance [54], etc., can detect nonequilibrium from different points of view.

Last but not least, as fundamental issues in cognitive neuroscience and neural information processing science, neural code [55,56] and neural communication [57] play an invaluable role in understanding the internal mechanisms of the brain and exploring certain medical conditions. In the complex nervous system, information is conveyed by spike trains, which are considered as elements of neural code. The exploration and detection of spike trains is a hot research field in neural code. For example, Mainen et al. [58] exploited recordings from neurons to detect the reliability of spike generation. Knoblauch et al. [59] examined the relationship between noise and inter-spike-interval statistics in real neurons in working brains. Pregowska [60] explored how spike fluctuations affect the information transmission rates. Moreover, how neurons communicate is also an important aspect of studying brain function. The frequencies and temporal dynamics of neural communication are associated with distinct behavioral states, and have an important impact on the flow of information in the brain [57]. Schizophrenia causes the abnormal function of neurons in the brain, which could exert an effect on neural code and communication [61,62]. The formation of brain networks based on MEG and the extraction of the network characteristics depend on the neural code and the communication between neurons. Consequently, the analysis of the brain MEG network for SCZs can provide powerful support for the exploration of the pathophysiological mechanisms of schizophrenia.

## 6. Conclusions

We examine the performance of three methods of Granger causality by model data test, and the results show that MKGC has a more satisfactory performance in characterizing nonlinear features. Then, MKGC is employed to construct the brain network of SCZs and HCs to explore the strength, nonequilibrium, and complexity of the SCZ MEG network. Compared with HCs, SCZs have decreased network strength, significantly increased nonequilibrium, and decreased complexity, suggesting that SCZ negatively affects brain network interactions. The network analysis based on MKGC could provide a reliable approach to identify the dynamic structure of the SCZ brain network and investigate the pathological and physiological mechanisms of SCZ.

Finally, we would like to emphasize that our findings have to be validated with a larger and more representative number of subjects, especially to check the performance of MKGC in the characterization of MEG-directed interaction networks.

## Figures and Tables

**Figure 1 entropy-25-01006-f001:**
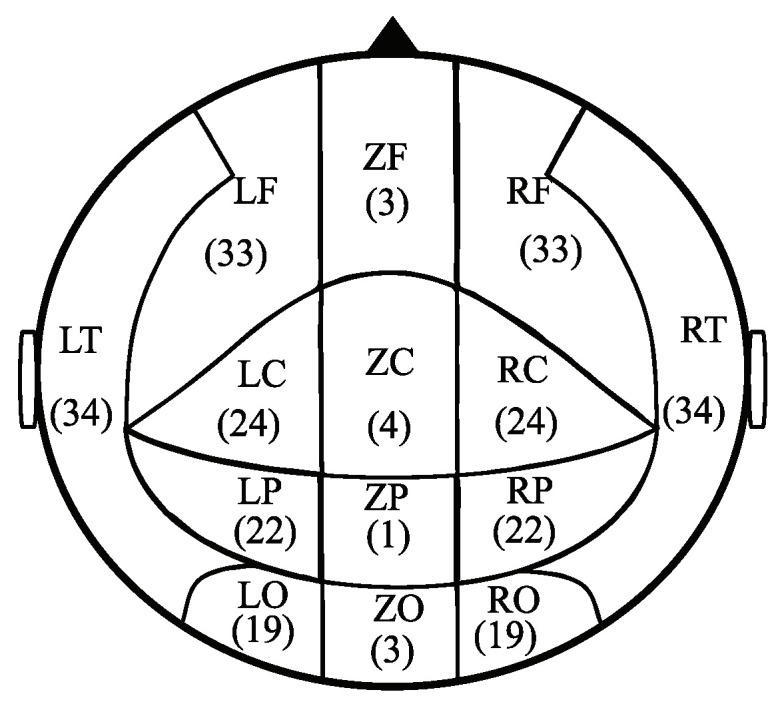
Layout of brain division of MEG recordings. Left frontal (LF), middle frontal (ZF), right frontal (RF), left central (LC), middle central (ZC), right central (RC), left temporal (LT), right temporal (RT), left parietal (LP), middle parietal (ZP), right parietal (RP), left occipital (LO), middle occipital (ZO), and right occipital (RO). The numbers represent the number of channels in each brain region.

**Figure 2 entropy-25-01006-f002:**
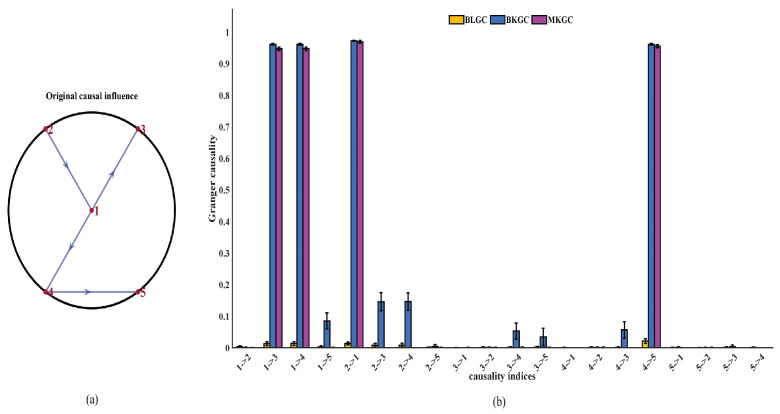
(**a**) Original causal influence of 1→3, 1→4, 2→1 and 4→5 between the five coupled autoregressive processes. (**b**) BLGC, BKGC and MKGC of the five coupled autoregressive processes. The order of autoregression is m=1 chosen by the Bayesian information criterion; MKGC and BKGC analyses are performed with the IP kernel (p=2); and vertical bars indicate estimated standard errors.

**Figure 3 entropy-25-01006-f003:**
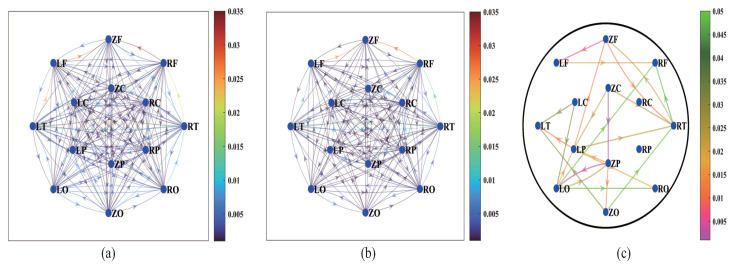
Effective connectivity network at group level for HCs (**a**) and SCZs (**b**). The nodes are the 14 brain regions. The colors of the links between nodes represent the interregional causal interactions, and the arrows indicate the directions of connections. (**c**) The directed differential connectivity graph between HCs and SCZs (p<0.05). The colors of the links represent the *p* values obtained by the Mann–Whitney U test.

**Figure 4 entropy-25-01006-f004:**
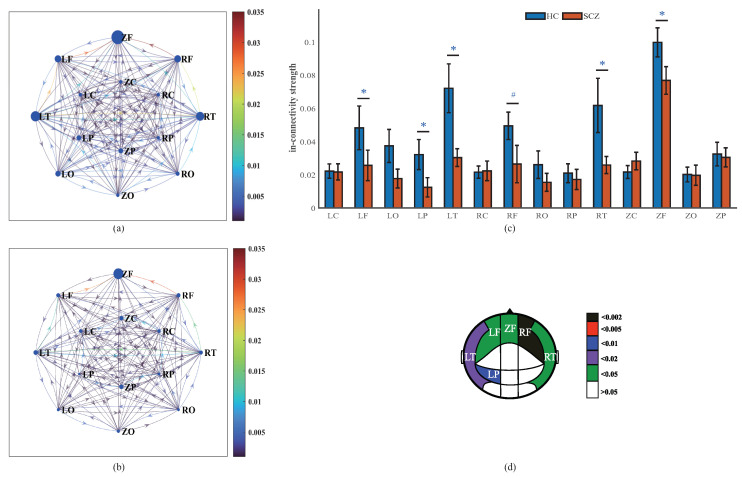
In-connectivity strength of brain regions. In-connectivity networks of HCs (**a**) and SCZs (**b**); the diameters of the nodes are positively related to the in-connectivity strength of the brain regions, and the colors of the links between nodes represent the causal interactions between the brain regions. (**c**) In-connectivity strength of brain regions (mean ± standard error); # and * indicate the statistical significance of p<0.002 and p<0.05 using the Mann–Whitney U test, respectively. (**d**) Brain regions with significant differences in in-connectivity strength. The fill color represents the *p* value obtained by the Mann–Whitney U test.

**Figure 5 entropy-25-01006-f005:**
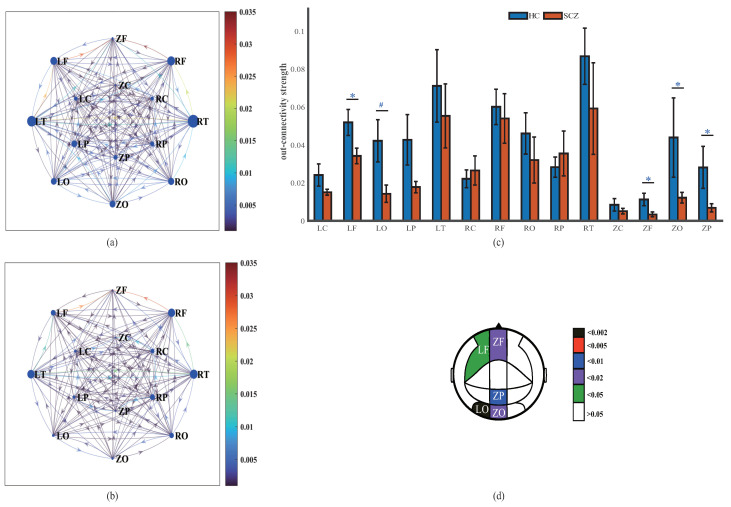
Out-connectivity strength of brain regions. Out-connectivity networks of HCs (**a**) and SCZs (**b**). The diameters of the nodes are positively related to the out-connectivity strengths of the brain regions and the colors of the links between nodes represent the interregional causal interactions. (**c**) The out-connectivity strength of brain regions (mean ± standard error); # and * indicate the statistical significance of p<0.002 and p<0.05 using the Mann–Whitney U test, respectively. (**d**) Brain regions with significant differences in out-connectivity strengths. The fill color represents the *p* value obtained by the Mann–Whitney U test.

**Figure 6 entropy-25-01006-f006:**
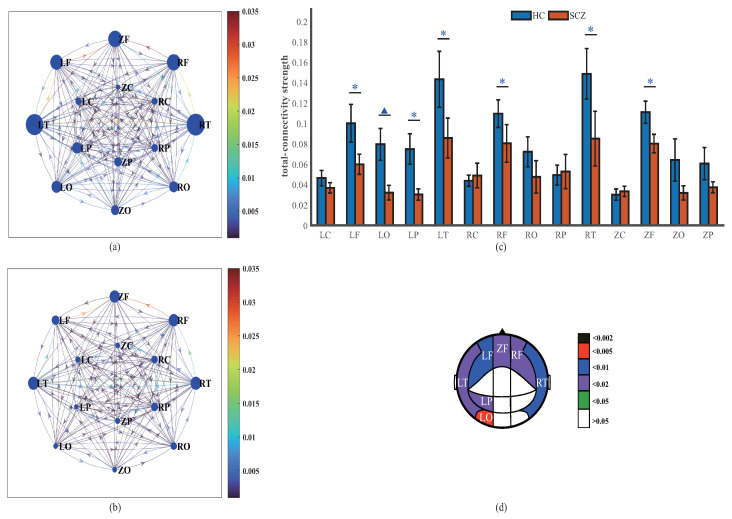
Total connectivity strength of brain regions. Total connectivity networks of HCs (**a**) and SCZs (**b**). The diameters of the nodes are positively related to the total connectivity strength of the brain regions, and the colors of the links between nodes represent the causal interactions between the brain regions. (**c**) Total connectivity strength of brain regions (mean ± standard error). △ and * indicate the statistical significance of p<0.005 and p<0.05 using the Mann–Whitney U test, respectively. (**d**) Brain regions with significant differences in total connectivity strength. The fill color represents the *p* value obtained by the Mann–Whitney U test.

**Figure 7 entropy-25-01006-f007:**
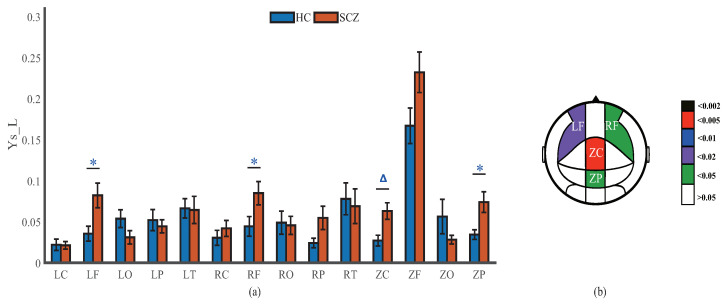
(**a**) Brain regional probabilistic difference $Y_{S_{-}}L$ of causal interactions (mean ± standard error). △ and * indicate the levels of significance (p<0.005 and p<0.05) of the probabilistic difference across groups using the Mann–Whitney U test. (**b**) Brain regions with significant differences in $Y_{S_{-}}L$. The fill color represents the *p* value obtained by the Mann–Whitney U test.

**Figure 8 entropy-25-01006-f008:**
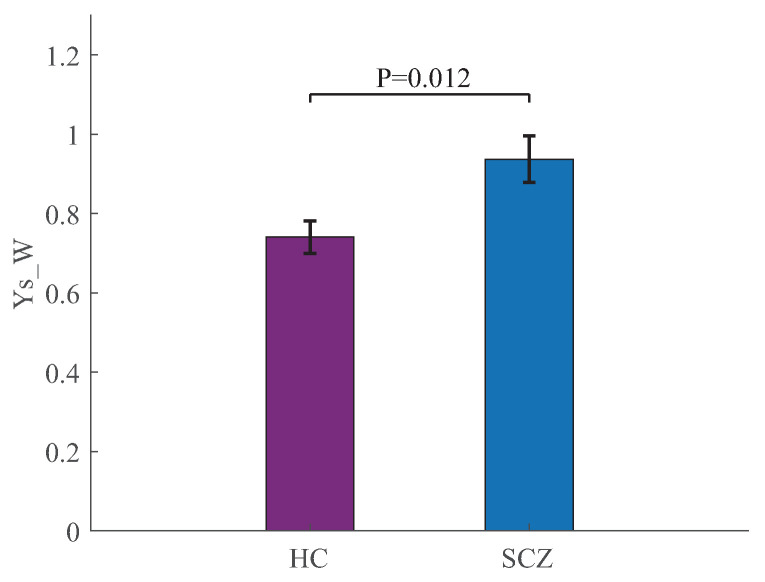
Nonequilibrium (mean ± standard error) of the MEG network constructed by MKGC over the whole brain for HCs and SCZs. The *p* value (p=0.012) is obtained by the Mann–Whitney U test.

**Figure 9 entropy-25-01006-f009:**
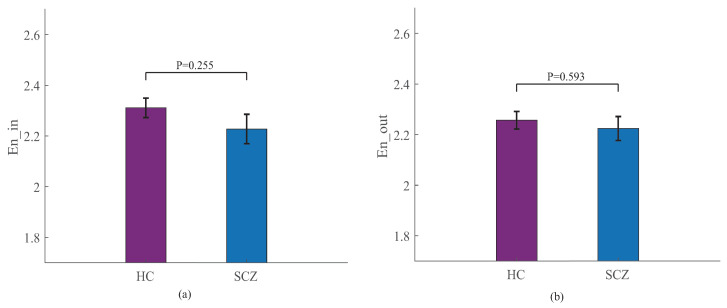
Shannon entropy of HCs’ and SCZs’ networks. (**a**) Shannon entropy of in-connectivity strength (mean ± standard error). (**b**) Shannon entropy of out-connectivity strength (mean ± standard error). *p* values are obtained by the Mann–Whitney U test.

**Table 1 entropy-25-01006-t001:** *Sensitivity*, *specificity*, and *Mcc* analysis of BLGC, BKGC, and MKGC.

Method	*Sensitivity*	*Specificity*	*Mcc*
MKGC	1 ± 0	0.9979 ± 0.0097	0.9945 ± 0.0260
BKGC	1 ± 0	0.5879 ± 0.0537	0.4329 ± 0.0402
BLGC	0.9653 ± 0.0866	0.8672 ± 0.0667	0.6890 ± 0.1369

## Data Availability

Restrictions apply to the availability of these data. Data was obtained from the Affiliated Brain Hospital of Nanjing Medical University and are available from Wei Yan and Jun Wang with the permission of the Affiliated Brain Hospital of Nanjing Medical University.

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
