# Peer review of "Schizophrenia MEG Network Analysis Based on Kernel Granger Causality"

_entropy, 2023, doi:10.3390/e25071006_

Round 1

Reviewer 1 Report

The Authors proposed multivariate kernel Granger causality and Shannon entropy to the analysis of schizophrenia magnetoencephalography.

I have the following comments and concerns: 

It is hard to obtain reliable (in statistical terms) results when analyzing only 31 patients, in particular a group of women (only 7 persons).

In my opinion, a brief paragraph about the fundamental problems in Neurosciences, specifically the problems concerning the form of neuronal codes (neural communication), because schizophrenia causes its disorders in a specific way, should be useful. This would also give some insight into why precise spike detection is so important. The Authors mentioned the problems with the good performance of the algorithms analyzed and developed for neurons with higher firing rates. The Readers of Electronics seem to be not very familiar with the meaning and questions related to neural code problems. Here are a few fundamental and recent papers that could be useful for the preparation of such paragraph:

van Hemmen, J. L.; Sejnowski, T., 23 Problems in Systems Neurosciences, Oxford University Press, Oxford, 2006.

Rieke, F.; Warland, D. D.; de Ruyter van Steveninck, R. R.; Bialek, W. Spikes: Exploring the Neural Code, MIT Press, 1997.

Pregowska, A. Signal Fluctuations and the Information Transmission Rates in Binary Communication Channels. Entropy 2021, 23, 92.

Mainen, Z. F.; Sejnowski, T. J., Reliability of spike timing in neocortical neurons. Science 1995, 268 (5216), 1503-1506.

Salinas, E.; Sejnowski, T. J., Correlated neuronal activity and the flow of neural information. Nature Reviews Neuroscience 2001, 2, 539-550.

Knoblauch, A.; Palm, G., What is signal and what is noise in the brain? Biosystems 2005, 79 (1–3), 83-90.

The Authors are based on Shannon's Information Theory and not citing his papers. In fact, the complexity is the entropy estimator. The definition of Shannon entropy should be recalled more precisely (formula 5).

Moreover, taking into account the specificity of both the journal and the paper, the abstract must be reformulated so that it translates the burden following the content of the paper into networks, and not into the description of the disease itself. Also, the quantitative value of the proposed algorithm/approach should be included.

At this moment I would recommend Major Revision.

The Authors proposed multivariate kernel Granger causality and Shannon entropy to the analysis of schizophrenia magnetoencephalography.

I have the following comments and concerns: 

It is hard to obtain reliable (in statistical terms) results when analyzing only 31 patients, in particular a group of women (only 7 persons).

In my opinion, a brief paragraph about the fundamental problems in Neurosciences, specifically the problems concerning the form of neuronal codes (neural communication), because schizophrenia causes its disorders in a specific way, should be useful. This would also give some insight into why precise spike detection is so important. The Authors mentioned the problems with the good performance of the algorithms analyzed and developed for neurons with higher firing rates. The Readers of Electronics seem to be not very familiar with the meaning and questions related to neural code problems. Here are a few fundamental and recent papers that could be useful for the preparation of such paragraph:

van Hemmen, J. L.; Sejnowski, T., 23 Problems in Systems Neurosciences, Oxford University Press, Oxford, 2006.

Rieke, F.; Warland, D. D.; de Ruyter van Steveninck, R. R.; Bialek, W. Spikes: Exploring the Neural Code, MIT Press, 1997.

Pregowska, A. Signal Fluctuations and the Information Transmission Rates in Binary Communication Channels. Entropy 2021, 23, 92.

Mainen, Z. F.; Sejnowski, T. J., Reliability of spike timing in neocortical neurons. Science 1995, 268 (5216), 1503-1506.

Salinas, E.; Sejnowski, T. J., Correlated neuronal activity and the flow of neural information. Nature Reviews Neuroscience 2001, 2, 539-550.

Knoblauch, A.; Palm, G., What is signal and what is noise in the brain? Biosystems 2005, 79 (1–3), 83-90.

The Authors are based on Shannon's Information Theory and not citing his papers. In fact, the complexity is the entropy estimator. The definition of Shannon entropy should be recalled more precisely (formula 5).

Moreover, taking into account the specificity of both the journal and the paper, the abstract must be reformulated so that it translates the burden following the content of the paper into networks, and not into the description of the disease itself. Also, the quantitative value of the proposed algorithm/approach should be included.

At this moment I would recommend Major Revision.

Author Response

Comment 1: It is hard to obtain reliable (in statistical terms) results when analyzing only 31 patients, in particular a group of women (only 7 persons).

Our response: We sincerely thank the reviewer for this valuable comment and we are sorry for the small dataset. 

From our current content on this small dataset, the results are effective in statistical analysis and could characterize the characteristics of schizophrenia effectively. 

In the revised version, we presented the discussion about the comparison of our results with some other works about schizophrenia network. The result of comparison indicated that our results are consistent with the findings of some other works about schizophrenia network analysis. Detailed contents are provided in Section 5 (page 12, lines 308-319). Additionally, we will be committed to expand a larger and more representative number of subjects to examine the performance of MKGC in characterizing MEG directed interaction networks.

Comment 2: In my opinion, a brief paragraph about the fundamental problems in Neurosciences, specifically the problems concerning the form of neuronal codes (neural communication), because schizophrenia causes its disorders in a specific way, should be useful. This would also give some insight into why precise spike detection is so important. The Authors mentioned the problems with the good performance of the algorithms analyzed and developed for neurons with higher firing rates. The Readers of Electronics seem to be not very familiar with the meaning and questions related to neural code problems. Here are a few fundamental and recent papers that could be useful for the preparation of such paragraph:

van Hemmen, J. L.; Sejnowski, T., 23 Problems in Systems Neurosciences, Oxford University Press, Oxford, 2006.

Rieke, F.; Warland, D. D.; de Ruyter van Steveninck, R. R.; Bialek, W. Spikes: Exploring the Neural Code, MIT Press, 1997.

Pregowska, A. Signal Fluctuations and the Information Transmission Rates in Binary Communication Channels. Entropy 2021, 23, 92.

Mainen, Z. F.; Sejnowski, T. J., Reliability of spike timing in neocortical neurons. Science 1995, 268 (5216), 1503-1506.

Salinas, E.; Sejnowski, T. J., Correlated neuronal activity and the flow of neural information. Nature Reviews Neuroscience 2001, 2, 539-550.

Knoblauch, A.; Palm, G., What is signal and what is noise in the brain? Biosystems 2005, 79 (1–3), 83-90.

Our response: Thanks for reminding us the lack of a brief paragraph about the fundamental problems in Neuroscience, specifically the problems concerning the form of neuronal codes (neural communication).

In the revised manuscript, We improved our discussion by citing the reviewer’s comments and the related papers. For detailed contents about the neural code and neural communication, please see Section 5 (pages 12-13, lines 347-364) and Refs. 56-61. We described the form and meanings of neural code and neural communication, and stated the important effects of neural code and neural interaction on information interactions in brain. We added the content to explain the reason why the brain  network analysis for MEG in patients of schizophrenia can provide a powerful tool for the exploration of the pathophysiological mechanism of schizophrenia from the perspective of neural code and neural communication.

Comment 3: The Authors are based on Shannon's Information Theory and not citing his papers. In fact, the complexity is the entropy estimator. The definition of Shannon entropy should be recalled more precisely (formula 5).

Our response: We sincerely thank the reviewer for this insightful and professional question about Shannon entropy.

In the revised manuscript, we added detailed information about Shannon's Information Theory, and cited the original paper (Ref. 32) and some important papers (Refs. 33-35,39) on Shannon entropy. We explained the definition of formula 5 with Shannon entropy clearly. Comprehensive contents are provided in Section 2.3.3 (page 5, lines 180-189).

References

32.Shannon, C.E. A mathematical theory of communication. The Bell system technical journal 1948, 27, 379–423.

33.Cover, T.M. Elements of information theory; John Wiley & Sons, 1999.

34.Xiong, W.; Faes, L.; Ivanov, P.C. Entropy measures, entropy estimators, and their performance in quantifying complex dynamics: Effects of artifacts, nonstationarity, and long-range correlations. Physical Review E 2017, 95, 062114.

35.Yao, W.; Yao, W.; Yao, D.; Guo, D.; Wang, J. Shannon entropy and quantitative time irreversibility for different and even contradictory aspects of complex systems. Applied Physics Letters 2020, 116, 014101.

39.Brissaud, J.B. The meanings of entropy. Entropy 2005, 7, 68–96.

Comment 4: Moreover, taking into account the specificity of both the journal and the paper, the abstract must be reformulated so that it translates the burden following the content of the paper into networks, and not into the description of the disease itself. Also, the quantitative value of the proposed algorithm/approach should be included.

Our response: Thanks for this valuable suggestion regarding the abstract in our manuscript.

In the revised manuscript, we reformulated the abstract carefully. We added some description of the quantitative values of the connectivity strength and the nonequilibrium feature and removed some descriptions about the disease. Detailed contents are provided in Abstract (page 1, lines 10-18).

Reviewer 2 Report

Using multivariate kernel Granger causality to construct directed weighted networks to characterize schizophrenia magnetoencephalography (MEG) is an interesting approach. It appears to work very well on the small dataset. The authors have caveact that the work is premised on a small sample size. But I am also hoping to see some comparison work with SZ MEG. The authors can improve the manuscript by strengthening the literature review and bringing in some comparison with SOTA.

-

Author Response

Comment 1: Using multivariate kernel Granger causality to construct directed weighted networks to characterize schizophrenia magnetoencephalography (MEG) is an interesting approach. It appears to work very well on the small dataset. The authors have caveact that the work is premised on a small sample size. But I am also hoping to see some comparison work with SZ MEG. The authors can improve the manuscript by strengthening the literature review and bringing in some comparison with SOTA.

Our response: We sincerely thank the reviewer for this insightful comment.

To strengthen reasonableness and effectiveness of our work, we first generate data based on coupled autoregressive processes to test the effectiveness of MKGC in comparison with the bivariate linear Granger causality and bivariate kernel Granger causality. Then, we apply MKGC to construct effective connectivity networks of MEG for patients with schizophrenia. Please see in Section 3 (pages 5-7).

Alternatively, we have done the followings in the revised manuscript to improve the reliability of the results more comprehensively.

We added some literature reviews about the findings of MEG network in schizophrenia in Section 1 (page 2, lines 35-47). We compared our results with some other existing works about schizophrenia network analysis, and the comparison manifested that our results are consistent with the findings of those works. Detailed contents are provided in Section 5 (page 12, lines 308-319).

Round 2

Reviewer 1 Report

The authors referred to my comments, indeed the proposed paper considers research on a very small group of people (they cannot be considered statistically), but Information Theory allows us to obtain indicative results on a smaller sample. I recommend the publication of this paper.

The paper is written in communicative English.

Reviewer 2 Report

Most concerns have been addressed.

Most concerns have been addressed.